# ScdNER: Span-Based Consistency-Aware Document-Level Named Entity Recognition

**Ying Wei**
Iowa State University / Ames, IA 50011
yingwei@iastate.edu

**Qi Li**
Iowa State University / Ames, IA 50011
qli@iastate.edu

## Abstract

Document-level NER approaches use global information via word-based key-value memory for accurate and consistent predictions. However, such global information on word level can introduce noise when the same word appears in different token sequences and has different labels. This work proposes a two-stage document-level NER model, ScdNER, for more accurate and consistent predictions via adaptive span-level global feature fusion. In the first stage, ScdNER trains a binary classifier to predict if a token sequence is an entity with a probability. Via a span-based key-value memory, the probabilities are further used to obtain the entity's global features with reduced impact of non-entity sequences. The second stage predicts the entity types using a gate mechanism to balance its local and global information, leading to adaptive global feature fusion. Experiments on benchmark datasets from scientific, biomedical, and general domains show the effectiveness of the proposed methods.

## 1 Introduction

Named entity recognition (NER) is an important task for many natural language processing applications (Lample et al., 2016; Li et al., 2020; Yu et al., 2020; Yan et al., 2019). Most NER models process sentences independently, leading to a risk of inconsistent prediction for the mentions of the same entity. That is, they can produce different predictions for the same entity mentions in different sentences depending on the contextual information for each sentence. Consideration of document-level features in entity representation learning is likely to obtain more consistent and accurate predictions since different occurrences of a specific token sequence within a document are likely to have the same entity types (Krishnan and Manning, 2006).

Based on this idea, some document-level NER (DL-NER) approaches (Luo et al., 2020; Gui et al., 2021) proposed to fuse global features of tokens via

token-based key-value memory, where the features of the same token are fused to produce document-level encoding for token label prediction. However, this token-level feature-sharing strategy may introduce noise in global feature fusing, as the same token occurrences in different contexts may not carry the same labels. For instance, labels for token "type" in the entity span "breast cancer type 1" and the token sequence "a unique type of cancer" are GENE-I and O, respectively (here, we call an entity span as a token sequence with an entity type). Fusing features of these two tokens will bring noise in predicting their labels.

To address the problem of token-level feature sharing, this work proposes a span-based consistency-aware DL-NER model (ScdNER) to enable context feature fusing on the span level. In particular, this work proposes a document-level context feature fusion technique for the same entity mentions. Given a document, we first extract document-level contextual encoding for each enumerated token sequence. Based on contextual encoding, we propose a two-stage prediction framework. In stage 1, a binary classifier is trained to predict if a token sequence is an entity span with a probability. Stage 2 employs a span-based key-value memory to enable context feature fusing of the same entity spans using probabilities from Stage 1 to reduce the impact of non-entity spans. The entity span features are updated by fusing global information with a gate to balance local and global features. Finally, the entity type is predicted for an entity span.

## 2 Related Work

DL-NER approaches use document-level contextualized information for more accurate and consistent predictions. Qian et al. (2019) use word sequence dependency structures to model non-local and non-sequential dependency. Akbik et al. (2019) propose to pool contextualized embeddings of each unique

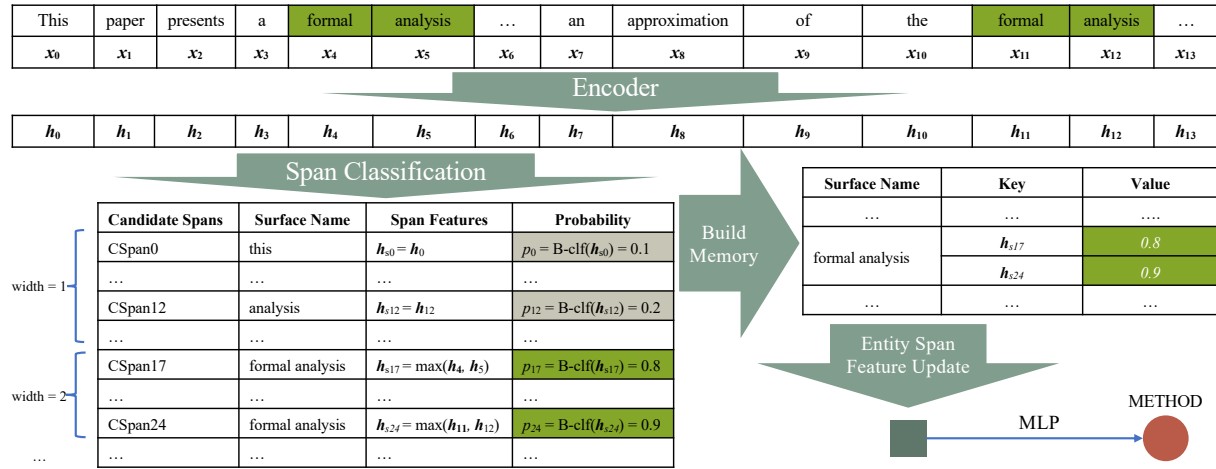

Figure 1: Illustration of the proposed ScdNER. In this example, "formal analysis" is a surface name shared by two entity spans. Their features are updated via fusing the other's contextual features, resulting in a "METHOD" prediction.

string. Luo et al. (2020) employ a word-based key-value memory to memorize word encoding, and an attention mechanism computes each token's document-level representations. Gui et al. (2021) design a two-stage label refinement network via a word-based key-value memory network. However, most existing works try to fuse word-level global information, which introduces noise when the same word in different entity spans has different labels. This work proposes to use a span-level key-value memory for global feature fusion.

## 3 ScdNER

**Problem Definition.** Given a document $\mathcal{D} = [w_1, \ldots, w_n]$ with $w_i$ representing the $i$th token, we enumerate all contiguous token sequences in each sentence as candidate spans $s_j = [w_a, \ldots, w_{a+k}]$ with $k \in \{0, 1, \ldots, K\}$, where $a$ is the starting token position in $s_j$ and $K$ is the maximum sequence length of spans. For each span, we aim to predict its label in $\{0, 1, \ldots, C\}$, where 0 indicates it is a non-entity and $C$ is the number of entity types.

**Our Model.** ScdNER is a two-stage DL-NER model. In the first stage, a binary span classifier is trained to predict if a span is an entity. The second stage uses a span-based key-value memory to adaptively fuse global features of spans with the same surface names for better span feature representation learning. Figure 1 illustrates the proposed ScdNER.

### 3.1 Document-Level Span Encoding

To encode candidate spans in the document, we first encode tokens $w_i$ with the document as context as follows. $\boldsymbol{H} = [\boldsymbol{h}_1, \ldots, \boldsymbol{h}_n] = \text{Encoder}([\boldsymbol{x}_1, \ldots, \boldsymbol{x}_n])$, where $\boldsymbol{x}_i \in R^{d_1}$ is the word embedding for $w_i$ and $\boldsymbol{h}_i$ is its contextual encoding. We can use a pre-trained language model such as BERT (Devlin et al., 2019) or LSTM model (Hochreiter and Schmidhuber, 1997) as an encoder. For a candidate span $s_j$, its encoding is computed by fusing token encoding:

$$\boldsymbol{h}_{s_j} = \text{AGG}(\boldsymbol{h}_a, \ldots, \boldsymbol{h}_{a+k}),$$

where $\text{AGG}(\cdot)$ is an aggregation function such as max-pooling and average-pooling. In the experimental studies, we use max-pooling as the aggregation function to introduce more non-linearity (Yu et al., 2014).

### 3.2 Stage 1: Binary Span Classification

In the first stage of ScdNER, we train a binary classifier to predict if a candidate span is an entity span. When building a span-based key-value memory, this binary classifier acts as a filter to reduce the noise in global feature fusion of the same candidate spans. For a candidate span $s_i$, the prediction is based on its encoding $\boldsymbol{h}_{s_i}$:

$$p_i = \sigma_1(\boldsymbol{W}_1 \boldsymbol{h}_{s_i} + \boldsymbol{b}_1), \qquad (1)$$

where $\boldsymbol{W}_1$ and $\boldsymbol{b}_1$ are trainable parameters and $\sigma_1(\cdot)$ is a sigmoid function. Using a threshold $\theta$, a candidate span $s_i$ is predicted as an entity span if its probability $p_i \geq \theta$. By span filtering, stage 1 outputs a list of entity spans $E = [e_1, e_2, \ldots, e_t]$. To train this binary span classifier, we create a binary label $y_i^b$ for each span $s_i$. If span $s_i$ is not an entity of any type, $y_i^b = 0$; otherwise, $y_i^b = 1$.

## 3.3 Stage 2: Adaptive Global Feature Fusion

We propose a span-based key-value memory to adaptively fuse features of the same entity spans.

**Span-based key-value memory.** The span-based key-value memory records representations of spans (keys) and their probabilities from Stage 1 (values). In particular, we create a document-level memory matrix $\boldsymbol{U} = [\boldsymbol{u}_1, \boldsymbol{u}_2, \ldots, \boldsymbol{u}_m]$, where the $i^{th}$ slot, $\boldsymbol{u}_i$, corresponds to span $s_i$ and stores a pair of vectors $(\boldsymbol{h}_{s_i}, p_i)$. Suppose $s_{q_1}, s_{q_2}, \ldots, s_{q_r}$ are spans under the same surface name $ph_j$. $ph_j$ can correspond to multiple slots, corresponding to a sub-array memory matrix $\boldsymbol{U}_{ph_j} = [\boldsymbol{u}_{q_1}, \boldsymbol{u}_{q_2}, \ldots, \boldsymbol{u}_{q_r}]$.

**Entity span feature update.** Based on the span-based key-value memory, each entity span $e_i$ updates its features to fuse global features of the same entity spans. If spans of the same surface name are identified as entities, they are likely to share the same semantic meanings. However, the assumption does not hold for non-entity spans. Thus, we use probabilities from Stage 1 to adjust the impacts of spans in global features to emphasize entity spans.

Given an entity span $e_i$, its features are updated by fusing features of other spans sharing the same surface name in the memory. To this end, we first extract its corresponding sub-array memory matrix $\boldsymbol{U}_{ph_j} = [(\boldsymbol{h}_{s_{q_1}}, p_{q_1}), \ldots, (\boldsymbol{h}_{s_{q_r}}, p_{q_r})]$. The entity span features are updated as:

$$\boldsymbol{h}^g_{ph_j} = \left( \sum_{k=1}^{r} p_{q_k} \boldsymbol{h}_{s_{q_k}} \right) / \sum_{k=1}^{r} p_{q_k}, \qquad (2)$$

$$g_i = \sigma_2(\boldsymbol{W}_2 \boldsymbol{h}_{e_i} + \boldsymbol{b}_2), \qquad (3)$$

$$\boldsymbol{h}'_{e_i} = g_i \boldsymbol{h}_{e_i} + (1 - g_i)\boldsymbol{h}^g_{ph_j}, \qquad (4)$$

where $\boldsymbol{W}_2, \boldsymbol{b}_2$ are trainable parameters, and $\sigma_2(\cdot)$ is a sigmoid function. In Eq. (2), the global features of a surface name $ph_j$ are computed by weighted average of all span features with their probabilities as weights. Thus, the impact of non-entity spans is reduced. Eq. (3) and Eq. (4) use a gate mechanism to balance the span's local and global features. If all spans of a surface name are predicted as non-entity, their features are not updated.

## 3.4 Entity Type Prediction Head

We use an entity type prediction head to predict the entity types for entity spans from stage 1. Following previous methods (Eberts and Ulges, 2020), we concatenate the entity span encoding and span width embedding, which leads to the final encoding for an entity span $e_i$ of width $t$: $\boldsymbol{f}_i = [\boldsymbol{h}'_{e_i}; \boldsymbol{d}_t]$,

where $\boldsymbol{d}_t$ is the width embedding of $t$. We use a multi-layer perceptron (MLP) to compute the prediction values $\boldsymbol{z}_i \in \mathcal{R}^C$ for all entity types, where $C$ is the number of entity types:

$$\boldsymbol{z}_i = \boldsymbol{W}_4 \sigma_3(\boldsymbol{W}_3 \boldsymbol{f}_i + \boldsymbol{b}_3) + \boldsymbol{b}_4, \qquad (5)$$

where $\boldsymbol{W}_3, \boldsymbol{W}_4, \boldsymbol{b}_3, \boldsymbol{b}_4$ are trainable parameters, and $\sigma_3$ is an element-wise activation function.

During prediction, a span $s_i$ will be predicted as non-entity if $p_i < \theta$. Otherwise, it will be assigned to entity type $c_i = \arg\max_j \boldsymbol{z}_i[j]$. It is worth noting that the value of $c_i$ can be 0, indicating that a span initially predicted as an entity span in Stage 1 can ultimately be classified as non-entity.

## 4 Experiments

We evaluate the proposed ScdNER model on benchmark NER datasets from various domains. We aim to study if span-level global feature fusing improves span feature learning and NER results.

**Experimental setups.** We use BioBERT (Lee et al., 2020), SciBERT (cased) (Beltagy et al., 2019), and BERT$_{base}$ (cased) (Devlin et al., 2019) based on Huggingface's Transformers (Wolf et al., 2019) as document encoder for biomedical, scientific, and general domain datasets, respectively. All hyper-parameters are tuned on the validation sets (See Appendix A for details). We report the average over 5 runs for ScdNER. **Evaluation metrics** include Precision, Recall, and F1 scores.

### 4.1 Experiments on Biomedical Datasets

**Dataset.** We use three biomedical datasets to evaluate ScdNER: CDR (Li et al., 2016), NCBI-disease (Doğan et al., 2014), and ChemdNER (Krallinger et al., 2015). The **CDR** dataset is tagged with 2 entity types and contains 500, 500 and 500 abstracts for train, validation, and test, respectively. The **NCBI-disease** dataset contains 593, 100, and 100 documents for train, validation, and test, respectively. It is tagged with 1 entity type: Disease. The **ChemdNER** dataset, tagged with 1 entity type, consists of 3,500, 3,500, and 3,000 abstracts for training, validation and test, respectively. More statistics can be found in the Appendix A.

**Baseline Models.** We compare ScdNER with SOTA biomedical NER models including BiLSTM-CRF (Luo et al., 2018), BioBERT, ConNER (Jeong and Kang, 2022), HierNER (Luo et al., 2020), DocLNER (Gui et al., 2021), and SpERT-Doc. SpERT-Doc is developed based on SpERT (Eberts and Ulges, 2020) using document-level encoding.

| Model | CDR | NCBI | ChemdNER |
|---|---|---|---|
| BiLSTM-CRF | - | 84.6 | 89.4 |
| BioBERT | 89.1 | 87.7 | - |
| ConNER | 89.9 | 89.1 | - |
| DocLNER | - | - | 90.7 |
| HierNER | - | - | 89.5 |
| SpERT-Doc | 90.1 | 88.8 | 92.1 |
| **ScdNER** | **91.2±0.07** | **90.4±0.32** | **93.1±0.11** |

Table 1: Results (F1 scores) on biomedical datasets.

| Model | P | R | F1 |
|---|---|---|---|
| DyGIE-GloVe | - | - | 65.20 |
| DyGIE++-BERT | - | - | 67.50 |
| PURE-SciBERT | - | - | 68.90 |
| SpERT-SciBERT-Sent | 70.87 | 69.79 | 70.33 |
| SpERT-SciBERT-Doc | 69.85 | 70.78 | 70.31 |
| PL-Marker-SciBERT | - | - | 69.90 |
| **ScdNER-SciBERT** | **71.60** | **71.43** | **71.44±0.20** |

Table 2: Results on SciERC dataset from the scientific domain. We report the Precision, Recall, and F1 scores.

| Model | Level | F1 |
|---|---|---|
| CVT-GloVe | Sent-level | 88.88 |
| DocLNER-GloVe | Doc-level | 88.49 |
| PoolNER-BERT | Sent-level | 89.71 |
| DocLNER-BERT | Doc-level | 90.28 |
| HierNER-BERT | Doc-level | 90.30 |
| SpERT-BERT | Sent-level | 90.24 |
| SpERT-Doc-BERT | Doc-level | 90.34 |
| **ScdNER-BERT** | Doc-level | **90.80±0.06** |

Table 3: Results (F1 scores) on OntoNotes 5.0 dataset.

| Model | P | R | F1 | #Type INC |
|---|---|---|---|---|
| SpERT-Doc | 89.7 | 90.5 | 90.1 | 80 (4.6%) |
| ScdNER | 90.6 | 91.3 | 90.9 | 58 (3.3%) |

Table 4: Prediction inconsistency study on the CDR dataset. We report the inconsistent rate of predictions.

**Main Results** are summarized in Table 1. ScdNER achieves consistently better performances than previous SOTA models on all datasets. ScdNER outperforms previous best models by margins of 1.1%, 1.3%, and 1.0% on CDR, NCBI-disease and ChemdNER, respectively, which demonstrates the effectiveness of conducting span-level global feature fusion in span feature learning.

### 4.2 Experiments on Scientific Datasets

**Dataset.** We evaluate ScdNER on SciERC (Luan et al., 2018) dataset, a scientific domain dataset constructed from abstracts of 500 AI papers, which are split into 350, 50, and 100 documents for the training, validation and test sets, respectively. The dataset is tagged with 6 scientific entity types.

**Baseline Models.** We compare with previous SOTA span-based joint entity/relation extraction models including DyGIE (Luan et al., 2019), Gy-GIE++ (Wadden et al., 2019), PURE (Zhong and Chen, 2021), SpERT, and PL-Marker (Ye et al., 2022). For fair comparisons, we use the same RE component and loss as SpERT during training.

**Main Results** are summarized in Table 2. From the results, the proposed ScdNER achieves consistently better performances than previous SOTA models on all metrics. In particular, ScdNER significantly outperforms previous models by at least 1.11% in terms of F1.

### 4.3 Experiments on General Domain Datasets

**Dataset.** We use the English portion of OntoNotes 5.0 dataset with gold-standard named entity annotations with 18 entity types. It consists of 2,483, 319, and 322 documents for the training, validation, and test sets, respectively.

**Baseline Models.** We compare with sentence-level NER models: CVT (Clark et al., 2018) and Pool-NER (Akbik et al., 2019), and document-level NER models: HierNER, DocLNER, and SpERT-Doc.

**Main Results.** We summarize the comparison results in Table 3. The proposed ScdNER outperforms the existing best model by a margin of 0.5% in terms of F1 score. Although entity repeating in the general domain is not as severe as in the biomedical domain, our method can still boost performance by better fusing global information. Error analysis of ScdNER is provided in Section 4.7.

### 4.4 Prediction Consistency Study

We study if ScdNER can address prediction inconsistency problem for entities of the same surface same on the CDR dataset. We use SpERT-Doc as baseline, which uses document-level context without considering label consistency. The results are summarized in Table 4. Compared to the baseline model, ScdNER can effectively reduce the rate of prediction inconsistency for entities of the same surface name, which shows the advantage of considering label consistency.

### 4.5 Ablation Study of ScdNER

We conduct ablation experiments on the NCBI-disease and SciERC datasets to study the contribution of each component in ScdNER. Based on ScdNER model, we remove key-value memory (stage 2) and entity filtering (stage 1) at a time. We also remove both to test the overall contribution of the adaptive global feature fusion module.

| Model | NCBI | SciERC |
|---|---|---|
| ScdNER | 90.39 | 71.44 |
| (-) Key-value memory | 89.28 | 70.36 |
| (-) Entity filter | 89.49 | 70.64 |
| (-) Both | 88.77 | 69.84 |

Table 5: Ablation study results on NCBI-disease and SciERC datasets. We report F1 (%) scores.

The ablation results are summarized in Table 5. We can observe that each component makes a significant contribution. In particular, when removing the span-level key-value memory, the performances drop by 1.11% and 1.08% on NCBI-disease and SciERC, which shows the benefit of conducting span-level global feature fusion for better span representation learning. When removing the entity filter while keeping the key-value memory, the performance drops by 0.90% and 0.80% on NCBI-disease and SciERC, which means the non-entity spans can introduce significant noise in adaptively fused global features, and it is necessary to reduce the impact of non-entity spans when fusing global information. Removing both leads to 1.52% and 1.60% performance drops on NCBI-disease and SciERC, respectively, demonstrating the overall contribution of the proposed methods.

## 4.6 Label Consistency on Datasets

The proposed methods are based on the assumption that the entities sharing the same surface name are highly likely to be assigned the same entity types. We examine this assumption on benchmark NER datasets from various domains. The statistics are summarized in Table 6. We calculate the number and rate of entities repeated in the document (shown at "#Entity REP" column) and the number and rate of entities with label inconsistency (shown at "#Type INC" column). We can observe high entity repeat rates and low type inconsistency rates, especially on biomedical NER datasets, which strongly supports the proposed methods.

## 4.7 Error Analysis

To better understand the bottleneck of ScdNER, we analyze the errors that ScdNER makes on the SciERC and OntoNotes datasets. We identify five major types of errors as follows: **1) Missing (M):** The model predicts no entity type for an entity span in the ground truth; **2) Extra (E):** The model predicts an entity type for a span that does not exist in the ground truth; **3) Wrong Boundary Correct Type (WC):** The model predicts the same entity type as that of an entity span in the ground truth with over-

| Dataset | #Entity | #Entity REP | #Type INC |
|---|---|---|---|
| CDR | 4,605 | 1,854 (40.3%) | 0 (0.0%) |
| NCBI Disease | 2,982 | 1,019 (34.2%) | 0 (0.0%) |
| ChemdNER | 15,911 | 5,345 (34.2%) | 0 (0.0%) |
| SciERC | 4,959 | 991 (19.9%) | 28 (5.6%) |
| OntoNotes | 58,177 | 10,766 (18.5%) | 173 (1.6%) |

Table 6: Label consistency statistics on benchmark datasets.

lapping spans; **4) Wrong Boundary Wrong Type (WW):** The model predicts a different entity type as that of an entity span in the ground truth with overlapping spans; **5) Correct Boundary Wrong Type (CW):** The model predicts a different entity type for an entity span in the ground truth.

The errors distribution of the ScdNER is shown in Table 8. Compared to the baseline model, ScdNER can significantly reduce the errors in "Extra" category. The proposed binary span classifier can greatly help with such kind of errors. Most errors of the ScdNER involve incorrect boundaries and incorrect entity type. The errors of incorrect boundaries come from these categories: Missing, Extra, Wrong Boundary Correct Type, and Wrong Boundary Wrong Type. The wrong entity type errors include Wrong Boundary Wrong Type and Correct Boundary Wrong Type. From the error distribution, the errors of incorrect boundaries are major and need to be addressed in the future. In particular, "Missing" and "Extra" contribute the most to the errors on both datasets. Table 10 in the appendix illustrates some examples.

## 5 Conclusion

This work proposes a two-stage DL-NER model, ScdNER, to enable more accurate and consistent NER prediction via global feature fusion of the same entity spans. In stage 1, ScdNER trains a binary span classifier to classify a span with a probability. Stage 2 uses a span-based key-value memory to adaptively fuse global features of the same entity spans. The probabilities in stage 1 are reused to reduce the impacts of non-entity spans. Empirical results on document-level NER datasets from various domains demonstrate the effectiveness.

## Acknowledgements

The work is supported in part by NIFA grant no. 2022-67015-36217 from the USDA National Institute of Food and Agriculture, and NSF 2237831 from the National Science Foundation.

## Limitations

In this work, we use SciERC (Luan et al., 2018) as one of the datasets to evaluate the proposed model. The kappa score for annotating entities in SciERC is only 76.9%, which indicates that only 35%-63% data are reliable (McHugh, 2012). Low-quality dataset can introduce noise and undermine the model performance. There is a chance that the performance gain from ScdNER model is due to over-fitting to the noise in annotation.

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

|          | P     | R     | F1    | AUC   |
|----------|-------|-------|-------|-------|
| SciERC   | 81.41 | 82.52 | 81.96 | 97.77 |
| OntoNotes| 91.32 | 93.82 | 92.55 | 99.71 |

Table 7: Evaluation of the binary classifier. We report Precision (%), Recall (%), F1 (%), and AUC scores.

## A  Experimental Setups

The threshold $\theta = 0.5$ is used in the binary span classifier. We use the AdamW optimizer (Loshchilov and Hutter, 2018) with a linear warmup. The learning rate starts from 5e-5 and is reduced by a linear decay learning rate schedule until 0. We mostly use the hyper-parameters in (Eberts and Ulges, 2020) with minor adjustments to better suit each dataset. We provide more hyper-parameters of ScdNER on five benchmark NER datasets in Table 9. We train all models using a Tesla V100 GPU. For the baseline model results we produced, we used the hyper-parameters reported in the original paper, albeit with slight fine-tuning.

## B  Performance Study of Binary Classifier

We use a binary classifier in the proposed ScdNER model to reduce the impact of noises and conduct first-stage prediction. The errors in this binary classifier may be propagated to the second stage, making a critical impact on the overall performance. We conduct experiments to study its effectiveness on the SciERC and OntoNotes 5.0 dataset, with results summarized in Table 7. From the results, we observe that the binary classifier achieves 81.96% and 92.55% F1 scores on two datasets, which is highly accurate. Thus, using binary classifier as an entity filter has the risk of error propagation, but can bring more benefits than errors from the results of ablation study. Future work can include soft punishment to avoid the error propagation issue.

| | SciERC | | OntoNotes | |
|---|---|---|---|---|
| | **ScdNER** | **SpERT** | **ScdNER** | **SpERT** |
| Missing | 188 | 197 | 350 | 265 |
| Extra | 142 | 205 | 473 | 632 |
| Wrong Boundary Correct Type | 120 | 103 | 327 | 322 |
| Wrong Boundary Wrong Type | 60 | 46 | 131 | 128 |
| Correct Boundary Wrong Type | 184 | 206 | 228 | 246 |

Table 8: Error distributions on SciERC and OntoNotes datasets.

| Dataset | Context | Pretrained LM | Batch Size | Learning Rate | Training Epoch | Threshold | Max Span Size $K$ |
|---|---|---|---|---|---|---|---|
| CDR | Document | BioBERT | 6 | 5e-5 | 30 | 0.5 | 8 |
| NCBI-disease | Document | BioBERT | 6 | 5e-5 | 30 | 0.5 | 10 |
| ChemdNER | Document | BioBERT | 6 | 5e-5 | 30 | 0.5 | 10 |
| SciERC | Document | SciBERT | 4 | 5e-5 | 30 | 0.5 | 10 |
| OntoNotes | Document | BERT | 4 | 5e-5 | 30 | 0.5 | 10 |

Table 9: Hyper-parameters of the named entity recognition settings.

| Category | Document |
|---|---|
| **M** | . . . Since we know that Jingguang Bridge is located in the CBD district, well there are many office buildings, ah, a lot of them, as well as quite a lot of friends who get up early in the morning to go to work. . . .

early in the morning: `TIME` |
| **E** | . . . In Taiwan, in recent years the Ministry of Economic Affairs, the National Science Council and the National Health Research Institutes have been strongly pursuing "biochip" research programs. . . .

recent years: `DATE` |
| **WC** | . . . Well, ah, if there is an accident, from the time it occurs until the time all traffic is cleared, it probably takes about half an hour to an hour during peak periods. . . .

about half an hour \| about half an hour to an hour: `TIME` |
| **WW** | However, southern North China and the Huang-Huai area will bid farewell to the snow tonight, ah. Tomorrow during the day, snowfall in the Jiang-Huai area will stop, and there will also be less snow in Northwest China. . . .

Tomorrow: `DATE` \| Tomorrow during the day: `TIME` |
| **CW** | . . . Judy Miller acknowledges that at Louis Libby's request this is the Dick Cheney aide she agreed to refer to him and in stories not as a senior administration official but as a former Hill aide because he had once worked on Capitol Hill. . . .

Capitol Hill: `ORG` \| `LOC` |

Table 10: Case study of NER results on the OntoNotes dataset. Here, green texts and blue texts indicate predictions with correct and wrong boundaries, respectively. The predicted entity types in `green box` and `blue box` indicate predictions with correct and wrong types, respectively.