# OpenReview forum: "ScdNER: Span-Based Consistency-Aware Document-Level Named Entity Recognition"
_EMNLP/2023/Conference — EMNLP 2023 Main_

### Official Review · Reviewer_G1FQ · 2023-07-31

**Soundness:** 3

**Excitement:**

3: Ambivalent: It has merits (e.g., it reports state-of-the-art results, the idea is nice), but there are key weaknesses (e.g., it describes incremental work), and it can significantly benefit from another round of revision. However, I won't object to accepting it if my co-reviewers champion it.

**Paper Topic And Main Contributions:**

The paper presents an approach for document-level NER, which aims to achieve consistent predictions for entities within a document. The approach is divided into two stages: first, a binary classifier identifies spans that mention entities. In the second stage, a key-value store is used to store and learn span information on a global, document-level information. Both local (the individual mention) and global representations are then used for prediction. Experiments on data sets from three different research areas (i.e., biomedical, scientific, and general domain) show superior results compared to competitor models.

**Questions For The Authors:**

- Q1: Concerning stage I of the approach: How is it determined in this context whether a span / string signifies an entity or not? Does it consider only the exact mentions from the gold standard, or does it also include substrings of these? (Lines 143, 144)

- Q2: Concerning state II of the approach: How do you handle prediction conflicts, e.g. if a span is classified as type A and a substring / subspan of this span is classified as type B?

- Q3: Concerning Appendix B: In this analysis, are you solely focusing on exact matches of entity spans, right? Wouldn't it be equally valuable for your research to examine how many entities include tokens that are also labeled as 'O' or as another entity (as exemplified by the "breast cancer type I" case from the introduction)? Have you conducted any examination in this regard?

- Q4: Have you ever explored the efficacy of an approach that initially employs a sentence-level classifier, and then in a post-processing step uniformly assigns the same labels to the same surface forms (e.g., if "breast cancer" is labeled as a disease in one sentence, label all other instances of "breast cancer" as a disease in the document)? This could potentially provide another baseline for your approach.

**Reasons To Accept:**

-	The work proposes (to the best of my knowledge) a novel approach for document-level named entity recognition.
-	The experimental results highlight the suitability of the proposed approach.
-	The article is linguistically essentially well written.

**Reasons To Reject:**

-	Important / major analyses to support the claim/motivation of the work as well as ablation studies for better result interpretation are only available in the appendix of the paper. Even with an additional page for publication the appendix’ content will not fit into the main text.
-	In the main text, only quantitative results are described, which makes it difficult to interpret the quality of the model and to assess whether and to what extent the intent of the approach can be realized.
-	Some details of the approach description are missing, which hampers reproducibility (see questions, improvements)

**Reproducibility:**

3: Could reproduce the results with some difficulty. The settings of parameters are underspecified or subjectively determined; the training/evaluation data are not widely available.

**Reviewer Confidence:**

4: Quite sure. I tried to check the important points carefully. It's unlikely, though conceivable, that I missed something that should affect my ratings.

**Typos Grammar Style And Presentation Improvements:**

-	The analyses in Appendix B and (at least one or two of) D, E, F should be included in the main text to strengthen argumentation. In my opinion, the work could have been made into a solid long paper.

-	It would also be interesting to see the results of the approach on biomedical datasets that have more than 2 entities (e.g. BioRED or BioNLP2013-ST-CG). Due to the higher number of entity types, there are also higher probabilities for inconsistent annotations.

---

> ### Author Rebuttal · Authors · 2023-08-28
>
> **Reasons To Reject**
>
> *Concern 1: Important/major analyses to support the claim/motivation of the work as well as ablation studies for better result interpretation, are only available in the appendix of the paper. Even with an additional page for publication, the appendix' content will not fit into the main text.*
>
> **Answer**
>
> Thanks for your feedback. We acknowledge that some parts in the appendix, including Sections B, C, E, and F, would be better to appear in the main text. To this end, we plan to do the following revisions such that these contents can fit into the five-page limit.
> -  We will only use the qualitative analysis part of Section C and leave Table 4 in the appendix.
> - Table 5 will be removed since most settings are shared among datasets with minor differences. We will describe these settings in the text.
> - The 2nd, 3rd, and 4th columns of Table 6 will be removed as this information is already included in the main text. This will streamline Table 6 into a single-column table. Initially, these columns were included in the table because there is no restriction on the page limit of the appendix.
> - We will reformat Table 7 to save space. For example, the text in Table 7 will be abbreviated.
> - Figure 1 will be simplified to save space.
>
> ---
>
> *Concern 2: In the main text, only quantitative results are described, which makes it difficult to interpret the quality of the model and to assess whether and to what extent the intent of the approach can be realized.*
>
> **Answer**
>
> Thanks for your suggestion. In the final version, we plan to incorporate Section C into the main text to display the qualitative results.
>
> ---
>
> *Concern 3: Some details of the approach description are missing, which hampers reproducibility (see questions, improvements)*
>
> **Answer**
>
> Thanks for your feedback. We will provide more details and release the source code once the paper is accepted.
>
> ---
>
> **Questions For The Authors:**
>
> *Q1: Concerning stage I of the approach: How is it determined in this context whether a span / string signifies an entity or not? Does it consider only the exact mentions from the gold standard, or does it also include substrings of these? (Lines 143, 144)*
>
> **Answer**
>
> Thank you for your valuable feedback. Our approach is in line with prior work, such as SpERT, and adheres to a strict matching criterion. In the benchmark NER dataset, each entity is annotated by the starting and ending positions of a contiguous token sequence within the document. In this context, a span is identified as an entity only if its starting and ending positions exactly align with those of a labeled entity. As for substrings, we treat them as separate spans that may share some tokens with the primary span, but their predictions operate independently. Importantly, the concept of a "subspan" is not applicable under the span-based NER settings.
>
> ---
>
> *Q2: Concerning state II of the approach: How do you handle prediction conflicts, e.g. if a span is classified as type A and a substring / subspan of this span is classified as type B?*
>
> **Answer**
>
> Essentially, the substrings are treated as separate spans that may share some tokens with the primary span. However, their predictions operate independently. For example, the span ``United States'' and its substring ``United'' are considered separate spans, and their predictions are made independently. In some benchmark datasets like the SciERC dataset, there are overlapping entity spans with different entity types and nested entity names. It means that the following cases occur in training data: a span is annotated as type A and a substring / subspan of this span is annotated as type B. And in inference time, the model needs to predict different labels for such cases, too.
>
> ---
>
> *Q3: Concerning Appendix B: In this analysis, are you solely focusing on exact matches of entity spans, right? Wouldn't it be equally valuable for your research to examine how many entities include tokens that are also labeled as 'O' or as another entity (as exemplified by the "breast cancer type I" case from the introduction)? Have you conducted any examination in this regard?*
>
> **Answer**
>
> Thank you for your constructive feedback. Yes, our focus is solely on the exact matching of entity spans. We operate under the assumption that only entity spans with identical surface names will have consistent labels. We do not make the same assumption for non-entity spans. If a span with a matching surface name is not annotated as an entity (and is thus labeled as "O"), its features will not be shared with other entity spans bearing the same surface name.
>
> In general, we don't think that the following assumption holds: if a span A shares the same surface name with an entity span B in the same document, then A must be an entity span with the same type of B. (To our understanding, this is the scenario your question concerns).
> Here is one example. The term "parsing" is labeled as "O" in this context: "After parsing the example image into its semantic components," yet it is labeled as a "Task" in the following context from the same document: "that utilizes recent advances in unification-based parsing."
>
> We do not want to consider the features of the "parsing" from the first context when building the key-value memory for entity "parsing" in the second context. In fact, fusing their features will introduce noise for entity spans.
>
> Since we only assume the label consistency of entity spans, analysis in Appendix B, which aims to validate this assumption, only concerns entity spans (i.e., spans that are not "O").
>
> ---
>
> *Q4: Have you ever explored the efficacy of an approach that initially employs a sentence-level classifier, and then in a post-processing step uniformly assigns the same labels to the same surface forms (e.g., if "breast cancer" is labeled as a disease in one sentence, label all other instances of "breast cancer" as a disease in the document)? This could potentially provide another baseline for your approach.*
>
> **Answer**
>
> Thank you for your thoughtful feedback. We have implemented the method you suggested, building upon SpERT, one of our baseline models. For predicted entities that share the same surface name, we assign them a uniform label based on the majority entity type. This method was evaluated using the SciERC dataset, and the comparative results are summarized in the subsequent table.
>
> | Model | Precision | Recall | F1 |
> | :---        |     :----:    |     :----:    |     :----:    |
> | SpERT | 70.87 | 69.79 | 70.33 |
> | SpERT + post | 69.89 | 71.19 | 70.53 |
> | **ScdNER** | 71.60 | 71.43 | **71.44** |
>
>
> Based on the results, it is shown that post-processing yields a marginal performance improvement, specifically a 0.2\% increase. We recognize the value of this as a solid baseline model and plan to include these findings in the final version of our paper.

---

### Official Review · Reviewer_3NNy · 2023-08-04

**Soundness:** 3

**Excitement:**

3: Ambivalent: It has merits (e.g., it reports state-of-the-art results, the idea is nice), but there are key weaknesses (e.g., it describes incremental work), and it can significantly benefit from another round of revision. However, I won't object to accepting it if my co-reviewers champion it.

**Missing References:**

1. “FLERT: Document-Level Features for Named Entity Recognition”, Schweter and Akbik,
2. “Exploiting Global Contextual Information for Document-level Named Entity Recognition”, Wang et al.


**Paper Topic And Main Contributions:**

The paper deals with the task of document-level named entity recognition. The goal is to ensure the same NER tags are being predicted across all mentions of an entity in a document. To this end, the paper proposed a two-stage approach. In the first stage, the model uses a binary classifier to identify the potential mention spans and stores the surfaces in a key-value memory. In the second stage, the model aggregates the span representations of each unique surface and applies a multi-class classifier MLP to predict the tags. Experiments are done on biomedical, scientific, and general domain NER data sets. The results outperform all baselines considered in this paper.

**Reasons To Accept:**

1. The study is done across a wide variety of domains and shows consistently better performance across all data sets.
2. The method is simple and effective.


**Reasons To Reject:**

1. The paper does not recognize the computational cost of the span-based approach as opposed to the sequence labeling approach. The complexity of the span-based approach is quadratic. So the binary classifier would need O(n*KC2) predictions. It is a limitation of any span-based approach. But I expected a discussion in the limitations section on this.

**Reproducibility:**

4: Could mostly reproduce the results, but there may be some variation because of sample variance or minor variations in their interpretation of the protocol or method.

**Reviewer Confidence:**

4: Quite sure. I tried to check the important points carefully. It's unlikely, though conceivable, that I missed something that should affect my ratings.

**Typos Grammar Style And Presentation Improvements:**

Section 3.3 is hard to read with so many notations and subscripts. Please try to simplify it.

---

> ### Author Rebuttal · Authors · 2023-08-28
>
> **Reasons To Reject**
>
> *The paper does not recognize the computational cost of the span-based approach as opposed to the sequence labeling approach. The complexity of the span-based approach is quadratic. So the binary classifier would need O(n*KC2) predictions. It is a limitation of any span-based approach. But I expected a discussion in the limitations section on this.*
>
> **Answer**
>
> Thanks for your feedback. It appears there might be some misunderstanding regarding the computational cost. The number of spans is linear to the length of the sentence. We've set a maximum sequence length for spans, denoted as $K$, as described in the main text (from lines 94 to 100). For a sentence with $n$ words, the total spans enumerated would be $(n + n-K+1) \times K / 2$, which is $O(nK)$. As $K$ is a constant, the overall computational overhead for the span-based technique is linear $O(n)$. In the experiments conducted in this work, we've employed $k = 8$ or $10$, as shown in Table 5.
>
> ---
>
> **Missing references**
>
> *1. "FLERT: Document-Level Features for Named Entity Recognition”, Schweter and Akbik,*
>
> *2. "Exploiting Global Contextual Information for Document-level Named Entity Recognition”, Wang et al.*
>
> **Answer**
>
> Thank you for your insightful suggestion. The two works you cited focus on encoding document-level features for NER tasks. The two baseline models aim to have each word's encoding contain global or document-level features. Specifically, FLERT explores the advantages of transformer-based document-level features in enhancing NER performance. These features are derived by processing a sentence in the context of its surrounding text to produce word embeddings enriched with document-level characteristics. Meanwhile, GCDoc captures global contextual information at both the word and sentence levels. At the word level, the document is transformed into a graph, and a Graph Neural Network is employed to encode the features of each node. At the sentence level, a specialized cross-sentence module is used to encode relationships between adjacent sentences.
>
> In contrast, our model enables each entity span's encoding to contain global features from other entity spans that share the same surface name. Our proposed model leverages a span-based key-value memory system to integrate the global features that are unique to each individual entity. We appreciate the relevance of the works you mentioned and will certainly cite them in the final version of our paper.
>
> ---
>
> **Typos Grammar Style And Presentation Improvements**
>
> *Section 3.3 is hard to read with so many notations and subscripts. Please try to simplify it.*
>
> **Answer**
>
> Thank you for your suggestion. We plan to streamline this section in the final version.

---

### Official Review · Reviewer_iSfy · 2023-08-07

**Soundness:** 4

**Excitement:**

3: Ambivalent: It has merits (e.g., it reports state-of-the-art results, the idea is nice), but there are key weaknesses (e.g., it describes incremental work), and it can significantly benefit from another round of revision. However, I won't object to accepting it if my co-reviewers champion it.

**Missing References:**

You should cite "Design Challenges and Misconceptions in Named Entity Recognition" since this paper is the first to suggest global features.

**Paper Topic And Main Contributions:**

This paper focuses on Document-Level NER, based on the intuition that spans with the same surface form are more likely to have the same entity type. In this paper, the authors propose a 2-stage DL-NER model. In stage 1, they train a binary classifier to classify whether each token is an entity or not. In stage 2, they build a span key-value memory to fuse the features of spans that share the same surface form. They evaluate their model on NER datasets from three domains: biomedical, scientific, and general domain. In these three domains, they show that their model is better than all the baseline models, including other document-level NER models, and also sentence-level NER models.

**Reasons To Accept:**

The paper is well written.
The performance is good. Evaluating on different datasets from three domains, and still outperforming.


**Reasons To Reject:**

Span-based key-value memory is updated by the matching of the surface form, which does not consider the situation of entity coreference.

**Reproducibility:**

4: Could mostly reproduce the results, but there may be some variation because of sample variance or minor variations in their interpretation of the protocol or method.

**Reviewer Confidence:**

3: Pretty sure, but there's a chance I missed something. Although I have a good feel for this area in general, I did not carefully check the paper's details, e.g., the math, experimental design, or novelty.

---

> ### Author Rebuttal · Authors · 2023-08-28
>
> **Reasons To Reject**
>
> *Q1. Span-based key-value memory is updated by the matching of the surface form, which does not consider the situation of entity coreference.*
>
> **Answer**
>
> Thank you for the feedback on the usage of entity coreference in our span-based key-value memory system.
>
> We agree on the importance of entity coreference. Currently, most NER benchmark datasets do not include coreference annotations to train coreference resolution models. In this work, the SciERC dataset serves as the sole data source containing coreference information. However, it's worth noting that the majority of state-of-the-art models do not leverage this coreference data. To ensure a fair comparison, we opt not to utilize entity coreference information. Instead, we consider entities with identical surface forms as co-referenced entities, thereby minimizing the potential for introducing noise into the model.
>
> For future work, we want to use the coreference information in SciERC and also establish new benchmark datasets with entity coreference annotations to explore the enhanced versions of the span-based key-value memory that can incorporate coreference resolution. We think the coreference resolution and name entity recognition could be modeled in a multi-task framework and help each other. We appreciate your invaluable feedback and see its potential implications for future advancements.
>
> ---
>
> **Missing References**
>
> *You should cite "Design Challenges and Misconceptions in Named Entity Recognition" since this paper is the first to suggest global features.*
>
> **Answer**
>
> Thanks for your suggestion. We will cite this paper in the final version.

---

### Meta-Review · Area_Chair_Zbk4 · 2023-09-14

**Recommendation:** 4

**Metareview:**

The paper presents a method for generating consistent NER labels across documents using the intuition that entities that share the same surface form are very likely to have the same NER label.

All three reviewers recognise the soundness of the approach and highlight the evaluation on multiple domains as an important contribution. Equally, all three reviewers are satisfied that the experiments demonstrate the increased performance of the proposed solution.

While any remaining technical issues raised by Reviewers 3NNy and G1FQ seem to have been addressed by the authors, the general consensus in the discussion was that the paper was rather incremental in nature which kept the reviewers from being more excited about it.  As the discussion with Reviewer G1FQ suggests, including some of the material currently found in the appendices would make this paper this paper more robust and the authors have proposed a plan to do so already.

---

### Decision · Program_Chairs · 2023-10-07

**Decision:**

Accept-Main

**Comment:**

The paper presents a method for generating consistent NER labels across documents using the intuition that entities that share the same surface form are very likely to have the same NER label.

All three reviewers recognise the soundness of the approach and highlight the evaluation on multiple domains as an important contribution. Equally, all three reviewers are satisfied that the experiments demonstrate the increased performance of the proposed solution.

While any remaining technical issues raised by Reviewers 3NNy and G1FQ seem to have been addressed by the authors, the general consensus in the discussion was that the paper was rather incremental in nature which kept the reviewers from being more excited about it.  As the discussion with Reviewer G1FQ suggests, including some of the material currently found in the appendices would make this paper this paper more robust and the authors have proposed a plan to do so already.